# The Effect of Nutrient Deficiencies on the Annual Yield and Root Growth of Summer Corn in a Double-Cropping System

**DOI:** 10.3390/plants13050682

**Published:** 2024-02-28

**Authors:** Chuangyun Wang, Yankun Ma, Rong Zhao, Zheng Sun, Xiaofen Wang, Fei Gao

**Affiliations:** 1College of Agronomy and Biotechnology, China Agricultural University, Beijing 100193, China; tb2020301001@cau.edu.cn; 2College of Agronomy, Shanxi Agricultural University, Taiyuan 030031, China; yankunma1218@163.com (Y.M.); zr1105x@163.com (R.Z.); sz15589262217@163.com (Z.S.)

**Keywords:** double-cropping system, nutrient deficiency, yield, root growth, sustainable agriculture

## Abstract

The North China Plain has a typical winter wheat–summer corn double-cropping pattern. The effects of nutrient deficiency conditions on the root characteristics and yield of summer corn in the double-cropping system were studied for four years. Long-term monotonous fertilization patterns undermine crop rotation systems and are detrimental to the sustainability of agricultural production. To complement the development of rational fertilization strategies by exploring the response of crop rotation systems to nutrient deficiencies, an experiment was conducted in a randomized complete block design consisting of five treatments with three replicates for each treatment: (1) an adequate supply of nitrogen and phosphate fertilizers and potash-deficient treatment (T1); (2) an adequate supply of nitrogen and potash fertilizers and phosphorus-deficient treatment (T2); (3) an adequate supply of phosphorus and potash fertilizers and nitrogen-deficient treatment (T3); (4) nutrient-sufficient treatment for crop growth (T4); and (5) no-fertilizer treatment (CK). The results showed that different nutrient treatments had significant effects on the root length density (RLD), root surface area density (RSAD), and root dry weight density (RDWD) in summer corn. At the physiological maturity stage (R6), the root indexes of RLD, RSAD, and RDWD were significantly higher in the 0–20 cm soil layer in T4 compared to CK, with an increase of 86.2%, 131.4%, and 100.0%, respectively. Similarly, in the 20–40 cm soil layer, the root indexes of T4 were 85.7%, 61.3%, and 50.0% higher than CK, with varied differences observed in the other nutrient-deficient treatments. However, there was no significant difference among the treatments in the 40–60 cm layer except for T4, whose root index showed a difference. The root fresh weight and root dry matter in T4, T3, T2, and T1 were increased to different degrees compared with CK. In addition, these differences in root indexes affected the annual yield of crops, which increased by 20.96%, 21.95%, and 8.14% in T4, T2, and T1, respectively, compared to CK. The spike number and the number of grains per spike of T4 were 10.8% and 8.3% higher than those of CK, which led to the differences in summer corn yields. The 1000-kernel weight of T4, T2, and T1 were 9.5%, 8.8%, and 7.4% higher than that of CK, whereas the determining nutrient was nitrogen fertilizer, and phosphorus fertilizer had a higher effect on yield than potassium fertilizer. This provides a theoretical basis for the effect of nutrient deficiency conditions on yield stability in a double-cropping system.

## 1. Introduction

Annual wheat–corn crop rotation is an essential cropping pattern in North China Plain, which can utilize the cumulative climatic conditions more efficiently, and reasonable rotation improves soil fertility and nutrient balance [1,2]. However, a long-term single fertilization pattern can harm the rotation system, leading to soil nutrient imbalance, mineral deficiency, and consequently reduced productivity [3]. At present, farmers of planting areas in the North China Plain often remove crop residues from the farmland, and the common fertilizer application is still based on composite fertilizers. Therefore, the soil’s long-term supply of nutrients for crop growth cannot be replenished, and this in the long run will lead to a reduction in the quality of crops in the rotational system, which is not conducive to the sustainability of agricultural production. The rationalization of fertilizer application strategies based on crop needs and soil nutrient status is essential for improving crop yields and minimizing environmental pollution, as well as for the long-term sustainability of production. A long-term study has shown that nutrient imbalances can have profound negative effects on soil fertility and quality and that such imbalances can lead to the disruption of soil structure, the imbalance of microbial communities, and nutrient loss [4,5]. However, unclear nutrient input rates and asynchrony between nutrient utilization and crop requirements at critical growth stages are major challenges in current nutrient management [6,7]. Understanding soil nutrient dynamics and crop response to nutrients is necessary to optimize nutrient management practices. Cropping systems that include crop residue incorporation and crop rotation with legumes have been shown to optimize soil fertility and alter soil nutrient cycling dynamics, and they are conducive to sustainable farming [8]. However, the impact of nutrient deficiencies on next season’s crop due to insufficient residual soil fertility in current crop rotation patterns is unclear due to residual soil fertility deficiencies resulting from long-term straw removal and compound fertilizer application by farmers in the North China Plain. Therefore, we explored the effects of nutrient deficiencies on an annual wheat–corn crop rotation system to complement the development of rational fertilizer application strategies.

As the primary organs of plants, roots play a crucial role in absorbing water and nutrients, providing mechanical support, and influencing the amount of dry matter in the aboveground part of the plant, which ultimately affects crop yield [9,10,11]. An important part of our research was studying the effects of deficient conditions on the root system of summer corn in a crop rotation system. Plant growth depends on the uptake of nutrient elements such as nitrogen, phosphorus, and potassium by the root system [12]. Most of the excess nitrogen in the soil is transferred through plant assimilation, substrate uptake, and other loss routes. Nitrate and ammonium nitrogen are the primary forms of inorganic nitrogen that most field crops obtain and are removed from the soil through the root uptake system [13,14]. Mineral nutrition is an important factor affecting plant root growth. Phosphorus regulates the growth and developmental status of crops and thus affects the yield; it not only serves as a major constituent of many important organic compounds that constitute the plant body but also participates in photosynthesis and related metabolic processes, etc., through different forms [15,16]. The application of potash fertilizer can significantly increase the content of potassium in the kernel, which is conducive to the photosynthesis of leaves and the regulation of osmotic potential, and it can also enhance the ability of the plant root system to absorb the nutrients needed for growth and thus facilitate the growth of the aboveground part of the corn and increase corn yield and its resistance to collapse [17,18,19]. Based on the results of crop root system response to various soil nutrients, more effective fertilizer transport strategies can be established.

Plants have high plasticity to concerning environmental factors [20,21,22]. Individuals of the same plant will have differences in plant size, growth rate, and the distribution of dry matter among different organs [23]. The high plasticity of plant roots plays a crucial role in the interaction between plants. The concept of rhizosphere regulation refers to the optimization of rhizosphere interaction for sustainable development through the regulation and management of each component in the rhizosphere ecosystem based on a deep understanding of rhizosphere processes [24,25]. Therefore, studying the root system can further clarify the interactions between plants and their host environment.

We conducted a study to assess the impact of nutrient-deficient conditions on annual yield, summer corn root morphology, and root distribution characteristics in a double-season winter wheat–summer corn rotation system. To analyze these characteristics, we compared the differences between treatment with single-nutrient deficiency and full-nutrient application. We aimed to evaluate the impact of nutrient deficiencies on field root characteristics and yield, describe the effects of individual nutrient deficiencies, and establish a theoretical basis for rational fertilization to improve soil texture and enhance root distribution. The main objectives of the current phase of the study are to (i) determine the effects of applying different nutrient deficiency treatments on annual yields in a crop rotation system, and (ii) determine the effects of applying different nutrients on corn root morphology and distribution. In addition, our subsequent studies will involve the quantitative exploration of nutrient deficiencies on residual soil fertility and nutrient acquisition by including subsequent crops in the rotation. Thus, the results of this study may be useful in determining the effects of nutrient deficiencies on yield and root traits in wheat–corn annual rotation systems and in developing effective nutrient management strategies to determine the specific effects of a single nutrient on an annual rotation system, which would be beneficial in maintaining crop productivity, soil health, and the sustainability of the rotation.

## 2. Materials and Methods

### 2.1. Experimental Design and Crop Management

The study site (35.67° N, 111.18° E; altitude of 445 m a.s.l.) was located in Bai Village, Xinjiang County, Yuncheng City, Shanxi Province, Central China. The area in which the experiment was situated possesses a temperate monsoon climate and four distinct seasons. The weather conditions during the growing season are shown in Figure 1. The soil type of the region is usually brown soil. The region is part of the Loess Plateau of China. The crop rotation of winter wheat and summer corn is the main agricultural production mode here. Before the planting season, topsoil was collected from each plot using a five-point sampling method, and samples were air-dried indoors, pulverized, and sieved; then, soil organic matter and major nutrient contents were determined. The soil base properties of the test site are shown in Table 1.

This experiment started in the winter wheat season in 2019. Three long-season winter wheat and summer corn plants were planted continuously. The experiment was replicated three times for each treatment, and a total of 15 plots were arranged in random blocks. The size of each experimental area was 30 m^2^. The treatments in this experiment were set up as an adequate supply of nitrogen and phosphate fertilizers and potash-deficient treatment (T1); an adequate supply of nitrogen and potash fertilizers and phosphorus-deficient treatment (T2); an adequate supply of phosphorus and potash fertilizers and nitrogen-deficient treatment (T3); nutrient-sufficient treatment for crop growth (T4); and no-fertilizer treatment (CK). In this experiment, the summer corn straw produced during the season was removed before sowing winter wheat, and all wheat straw in the field was removed before sowing summer corn as a means of avoiding nutrient accumulation due to straw incorporation. The farmland was rototilled with a rotary tiller, and the summer corn and winter wheat were sown by hand after a compaction operation, with the same agronomic measures applied to all treatments. Throughout the growing season, urea fertilizer (600 kg/ha) with 46% nitrogen (N) content, calcium superphosphate (450 kg/ha) (P_2_O_5_ = 15%), and potassium sulfate (300 kg/ha) (K_2_O = 50%) were used as basal fertilizers before sowing wheat. Winter wheat was sown on 15–20 October 2019–2021, and summer corn was sown on 20–25 June 2021–2022. The winter wheat variety was Malan No. 1, and the summer corn variety was Denghai 605. Winter wheat was harvested on 10–15 June for all 2019–2021 plots, and summer corn was harvested on October 10–15 for all 2020–2022 plots. Hand weeding removal was conducted during each planting season.

### 2.2. Sampling and Measurements

#### 2.2.1. Yield

Crop yield measurements were conducted in the center row of each plot. Summer corn yield was measured at the physiological maturity stage (R6), where thirty consecutive plants per row per plot were harvested and counted. The yield components include the number of grains per ear (GN), the number of ears per hectare (EN), and the thousand-kernel weight (TKW). Corn kernel yield was recorded at 14% moisture content.

#### 2.2.2. Root Morphology

Root samples were obtained from three corn plants from each treatment at two critical reproductive stages: the tasseling stage (VT) and the physiological maturity stage (R6). Corn roots were collected in three depths, with each 20 cm layer, for a total sampling depth of 60 cm. To collect the underground parts, half the distance between the rows of corn and half the distance between the plants were chosen as the area from which to collect roots, and the volume of soil used was calculated to be 29,400 cm^3^. A hole was dug next to the demarcated area to observe the depth of the soil. After passing the soil through a 0.5 mm sieve in each layer, the root sample was cleaned using a hose with a spray nozzle attachment and then collected into a plastic bag for storage [26]. Indicator records included root fresh weight (RFW), root length, and root diameter for the samples taken. Afterward, the roots from each layer were then placed in a clear glass dish (25 × 35 cm), filled with a small amount of water to make the scanning image more translucent, and scanned with an HP Scanjet 8200 (Hewlett-Packard, Palo Alto, CA, USA) as well as with an image analyzer (Delta-T Area Meter Type AMB2; Delta-T Devices, Cambridge, UK) to measure metrics such as root length.

The root length density (RLD) is calculated by dividing the root length of a sample by the soil volume, and the root length measurement is expressed as density per unit volume as follows:$$\mathrm{RLD}\left({\mathrm{mm}\;\mathrm{cm}}^{-3}\right)=\frac{\mathrm{Root}\;\mathrm{length}\;(\mathrm{mm})}{{\mathrm{soil}\;\mathrm{volume}\;(\mathrm{cm}}^{3})}$$

The root surface area density (RSAD) is calculated as the root surface area divided by the sampled soil volume:$$\mathrm{RSAD}\left({\mathrm{mm}}^{2}{\;\mathrm{cm}}^{-3}\right)=\frac{{\mathrm{Root}\;\mathrm{surface}\;\mathrm{area}\;(\mathrm{mm}}^{2})}{{\mathrm{soil}\;\mathrm{volume}\;(\mathrm{cm}}^{3})}$$

The root dry weight density (RDWD) is calculated by dividing the sampled soil volume by the root dry weight (RDW):$$\mathrm{RDWD}\left({g\;\mathrm{dm}}^{-3}\right)=\frac{\mathrm{Root}\;\mathrm{dry}\;\mathrm{weight}\;(g)}{{\mathrm{soil}\;\mathrm{volume}\;(\mathrm{cm}}^{3})}$$

### 2.3. Statistical Analysis

The data for all planting seasons were analyzed using Microsoft Excel 2019 (Microsoft Corp., Redmond, WA, USA), Sigma Plot 11.0 (Systat Software, San Jose, CA, USA), and SPSS 26.0 (SPSS Inc., Chicago, IL, USA). Analysis of variance was used to verify the significant differences among the means, followed by Duncan’s multiple range test.

## 3. Results

### 3.1. Dry Matter Accumulation

Fertilizer application had a remarkable effect on the dry matter accumulation of summer corn at the R6. The accumulation of dry matter in the aboveground portion of corn was higher in all fertilizer treatments than in CK; it was 78.1%, 39.6%, 39.1%, and 53.6% higher in T1, T3, T2, and T4, respectively (Figure 2).

### 3.2. Root Fresh Matter and Root Dry Matter

Different nutrient fertilization treatments affected the root fresh matter and dry matter of summer corn roots. At the VT, the root fresh matter of T4, T3, T2, and T1 was 110.3%, 20.1%, 41.3%, and 46.2% higher than that of CK, respectively. The root dry matter of T4, T3, T2, and T1 were 88.9%, 25.9%, 22.2%, and 11.1% higher than that of CK, respectively. At the R6, the root fresh matter of T4, T3, and T1 was 81.7%, 31.0%, and 43.0% higher than that of CK, respectively. The root fresh matter did not differ in T2 compared with CK. The root dry matter of T1, T2, T3, and T4 were 14.3%, 28.6%, 19.0%, and 33.3% higher than that of CK, respectively (Figure 3).

### 3.3. Root Characteristics of Summer Corn

#### 3.3.1. Root Length Density, Root Surface Area Density, and Root Dry Weight Density

Fertilizer formulation had remarkable effects on summer corn root indicators such as root length density (RLD), root surface area density (RSAD), and root dry weight density (RDWD) at both stages. At the VT, the root indicators RLD, RSAD, and RDWD were 128.6%, 109.0%, and 116.3% higher in the topsoil layer (0–20 cm) of T4 than in CK. The indicators were, respectively, 71.4%, 66.5%, and 104.7% higher in T2 than in CK in the 0–20 cm soil layer. The T4 treatment resulted in 22.2%, 60.6%, and 44.2% increases in RLD, RSAD, and RDWD, respectively, in the topsoil layer. At the R6, root indicators such as RLD, RSAD, and RDWD of T4 were increased by 86.2%, 131.4%, and 100.0%, respectively, in the topsoil layer compared to the CK. The individual indicators of T3 in the topsoil layer were higher by 41.4%, 43.0%, and 20.0%, respectively, compared to the CK. The root indicators such as RLD, RSAD, and RDWD were increased by 24.1%, 64.5%, and 35.0%, respectively, in the topsoil layer in T2 compared to CK. Similarly, T1 also showed an increase of 87.9%, 115.7%, and 71.7% in various indicators in the topsoil layer compared to the CK treatment (Table 2).

Root indicators such as RLD, RSAD, and RDWD differed among fertilizer application treatments in the intermediate soil layer (20–40 cm). At the VT, the root indicators such as RLD, RSAD, and RDWD of T4 were, respectively, 60.0%, 70.0%, and 12.5% higher than those of CK in the intermediate soil layer. The indicators of T3 in this layer were 40.0%, 20.0%, and 32.5% higher than those of CK. The indicators of T2 in the 20–40 cm layer were 24.0%, 14.0%, and 7.5% higher than those of CK. In the intermediate soil layer, T1 increased by 40.0%, 26.0%, and 75.0%, respectively, compared with CK in all indicators. At the R6, the root indicators such as RLD, RSAD, and RDWD were increased by 85.7%, 61.3%, and 50.0%, respectively, for T4 compared to CK in the 20–40 cm soil layer. The T3 treatment resulted in 71.4%, 69.9%, and 33.3% higher indicators in the intermediate soil layer, respectively. The indicators of T2 were 6.5%, 12.3%, and 16.7% higher than those of CK in this soil layer. The root indicators of T1 were 16.9%, 68.1%, and 33.3% higher than CK, respectively. In the deep soil layer (40–60 cm), only the T4 root indicators showed differences, with no obvious differences between the other treatments (Table 2).

#### 3.3.2. Root Diameter and Root Length

Fertilizer application significantly affected root diameter and root length for summer corn. At the VT, the root diameter and root length of summer corn in the topsoil layer were remarkably different among treatments, and the T4 treatment increased the root diameter and root length by 96.0% and 136.0% in the topsoil layer, respectively, compared with CK. The root indicators such as root diameter and root length increased by 66.0% and 76.0%, respectively, in the topsoil layer for T2. The T1 treatment increased the root diameter and root length in this soil layer by 15.0% and 60.0%, respectively, compared to CK. At the R6, compared with CK, the root diameter and root length increased by 77.0% and 89.0% higher in the topsoil layer for T4. The T3 treatment increased the root diameter and root length in the topsoil layer by 20.0% and 42.0%, respectively. The T2 treatment increased the root diameter and root length by 39.0% and 24.0%, respectively, in the topsoil layer. The T1 treatment increased the root diameter and root length in the topsoil layer by 72.0% and 57.0%, respectively, compared to CK (Table 3).

In addition, the root diameter and root length were higher in T2 than in CK in the intermediate soil and deep soil layer (20–60 cm), whereas no significant differences were observed in the root diameter and root length in the other treatments in the deep soil layer (40–60 cm).

### 3.4. Grain Yield

The application of different fertilizers had a significant effect on the grain yield of summer corn. The grain yield was 32.2%, 21.0%, and 16.9% higher in T4, T2, and T1, respectively, than in CK. The grain yield of T3 did not differ from that of CK. The annual yield of T4, T2, and T1 increased by 20.96%, 21.95%, and 8.14% compared to CK. The spike number and the number of grains per spike of T4 were 10.8% and 8.3% higher than those of CK, which led to differences in grain yield. The 1000-kernel weight of T4, T2, and T1 were 9.5%, 8.8%, and 7.4% higher than that of CK (Table 4).

### 3.5. Correlation Analysis of Yield and Root Characteristics

The root length, root diameter, root volume, effective root area, effective absorptive area, RFW, and RDW of the underground part of summer corn at VT and R6 were remarkably correlated with its yield. The results showed that the root length, root fresh weight, and root dry weight had a higher correlation with summer corn yield, while the root fresh weight had no significant correlation with the root length and root diameter. These results suggest that the grain yield of summer corn is strongly influenced by root characteristics, which are affected by fertilizer management practices (Table 5).

## 4. Discussion

The growth and development of the root system of a plant is genetically controlled but is also influenced by changes in environmental factors; meanwhile, the root system acts as a link between the environment and crops, allowing them to absorb and utilize resources from the surrounding environment [27,28]. Appropriate nitrogen and essential trace mineral nutrients are critical for crop growth and development, and the application of the right amount of nutrients promotes vigorous root growth [29,30]. The lack of certain elements can cause metabolic impairment in crops, affecting root growth and crop development.

Plant root systems are sensitive to environmental nutrient responses, so root system indicators can be used to characterize the effects of nutrient deficiencies on plants. Typically, the total weight of the root system is only 10–20% of the plant, but it has a major impact on the dry matter content of the aboveground portion of the plant and crop yield [31], and the biomass of the corn root system is correlated with active uptake. In addition, in this study, nutrient deficiencies significantly affected crop dry matter quality, for both above- and belowground parts, and nitrogen, phosphorus, and potassium supply deficiencies all reduced the absolute biomass. These results are similar to the conclusions of a previous study in which nutrient deficiencies led to crop root damage and biomass reduction [32,33]. Moreover, root system development also has a significant correlation with grain yield. In this study, compared with CK, T3 resulted in a lower grain yield, while the grain yield of T1 did not differ from that of T2. Different nutrient-deficient treatments showed different grain yields, with T3 having similar or even lower yields than CK, whereas there was no significant difference in grain yield between T2 and T1. These results lead to similar conclusions to previously conducted research [34,35] in which nutrient deficiencies resulted in impaired root development and thus induced losses in crop yield.

The root system plays an important role in regulating crop growth, and sensory mechanisms at the root tip monitor the nutrients contained in the soil. This information contributes to the formation of chemical signals for root growth [36,37]. The root length and root diameter are the main structural features of the root system, and crops can adapt to their environment by adjusting morphological features such as roots and stems. The root characteristics of summer corn are the main factors that affect crop nutrient and water use efficiency [26]. In the present study, the distribution of summer corn roots in the soil was altered by the different nutrient allocations. In the current study, all other treatments led to different yields compared to CK. T4, T1, T3, and T2 showed a significant increment in the total root length and surface area in the 0–40 cm soil layer, which was not observed in the 40–60 cm soil layer except for T4. Hence, the most beneficial strategy adopted by plant growth appears to be the increase in total nutrient uptake through nutrient foraging strategies in the surface soil rather than root exploration in deeper soil layers.

The key indicators for studying root growth dynamics and responses to environmental conditions are the root length density (RLD), the root surface area density (RSAD), and the root dry weight density (RDWD) [38]. Also, the growth and spatial distribution of the root system in the belowground part of the crop determined its capacity to absorb nutrients and water, which, in turn, had a considerable impact on the growth of the aboveground part of the crop and the formation of the ultimate yield. Our research revealed that the RLD, RSAD, and RDWD were significantly higher in T3, T2, and T1 treatments at the VT and R6, and were highest in T4 at both reproductive periods, compared to CK. These results indicate that the soil application of adequate nitrogen, phosphorus, and potassium can change root traits. As a result of increased mineralization and root activity in the soil, the amount of available nitrogen increased, which in turn led to a more efficient uptake of nutrients from the soil by the plant, promoting plant growth and development and increasing crop dry matter accumulation and yield. Although all nutrient deficiency treatments promoted summer corn root development, they did so to varying degrees. The key factors for achieving high yields in corn are a strong nutrient uptake capacity and adequate nutrient supply at the growth and developmental stages [39]. The vigor and physiological activity of the root system can directly affect the plant’s vital activities, and increasing the root uptake area and improving its effectiveness can enhance the root–soil contact area [40]. In the present study, we found that the root samples undergoing T4 treatment had a higher effective root area and effective absorptive area than the CK, which ensured that they thrived throughout the grain-filling stage.

Our study showed that different types of nutrient deficiencies significantly affected root morphology and consequently root function. We compared the results of the differences in root system indexes (RDWD and RSAD) between treatments with adequate nutrient supply (T4) and nutrient-deficient treatments (T1, T2, and T3) at the two stages. By examining the differences between the nutrient-deficient treatments and T4, we can better visualize the effect of a single nutrient deficiency on these indicators, bearing in mind that the greater the difference between the two values of an indicator, the greater the effect of that nutrient type on that indicator. The results showed that the nutrient deficiency included in the T2 treatment had a greater effect on the RDWD, while the RSAD was more affected by the T2 and T3 nutrient allocation treatments (Figure 4).

Different types of nutrient deficiencies led to significant differences in crop root growth. The root length and root length density were much lower under nitrogen deficiency than under adequate nitrogen supply and led to a significant reduction in each indicator at both reproductive periods. However, it has also been found that nitrogen deficiency stimulates root growth in the early stages of corn fertility and reaches a maximum before stamen withdrawal [41,42]. We found that potassium deficiency significantly led to a reduction in the root length or root length density, but the difference in the root diameter was smaller in potassium-deficient treatments compared to T4, suggesting that potassium nutrient deficiency did not significantly affect the root diameter, which is in line with the findings of previous studies in which there was no difference in the mean root diameter of corn across different potassium-deficient treatments [43,44]. Most studies noted smaller root lengths and RLD values under low-potassium conditions, but the results were not significant and were more pronounced in deeper soils. Zhao et al. (2016) [17] observed that the mean root diameter decreased under low-potassium conditions, which is the same as our findings according to which the mean rhizomes tended to be smaller under potassium-deficient conditions.

Correlation analysis revealed that crop yield was remarkably and positively related to root system indicators such as the crop root length density, absorbed area, dry weight, and volume at two critical fertility stages, VT and R6. All these results also indicate that root characteristics influenced by various fertilizer applications are important factors in improving grain yield. In the future, we aim to investigate the modulation of root morphology through fertilization strategies to ultimately achieve some degree of yield modulation.

This paper focuses on the effects of the main types of nutrient deficiencies in summer corn production in the North China Plain, as well as the biomass and structural characteristics of crop roots under different deficiency types. In future research, we plan to refine the theory of nutrient deficiency effects accordingly. However, there are still some limitations. Some other studies possess data on the root–shoot ratio, specific root morphology, and other indicators, which we will refer to in the future to explore the mechanism underlying the optimization of the nutrient deficiency regulation of root morphology and understand the deficiency response more concretely. Climatic factors, combined with experimental results, revealed that the dry matter weight and root fresh weight metrics of summer corn had larger values in 2022, the year of abundant precipitation, while the root length and root diameter metrics had the opposite values, which could be attributed to the lack of precipitation, leading to the occurrence of stressful conditions [45,46], which contributed to the larger root length. Due to the large difference in precipitation data between the two years, we finally used the average value as the experimental result to minimize the interannual error. In this study, we categorized treatments without the application of a nutrient (0 kg/ha) as “deficient”, but it is possible that control treatments were not affected by nutrient deficiency. In addition, our future research will focus on the relevant interactions that may affect crop morphology, and nutrient deficiencies may also modulate root morphology and microbial communities, which remain open questions for further research.

## 5. Conclusions

Compared to the CK without fertilizer, the T4, T2, and T1 treatments with fertilizers all resulted in higher summer corn yields, with significant increases of 32.2%, 21.0%, and 16.9%, respectively. The root indicators such as the root length, root volume, root length density, and root surface area density were enhanced in T4 and T2 treatments compared to the CK treatment. These effects on root characteristics may explain the increase in crop yield. Some root characteristics were significantly more augmented by the T2 treatment than by the T1 treatment. This shows that nitrogen fertilizer significantly affects crop yield and root morphology. In addition, crop root development and growth were more affected by phosphorus fertilizer application than by potash fertilizer application. Based on the results of our study, we recommend a moderate increase in phosphorus fertilizer application to wheat–corn rotation systems to ensure that summer corn maintains a healthy root system.

## Figures and Tables

**Figure 1 plants-13-00682-f001:**
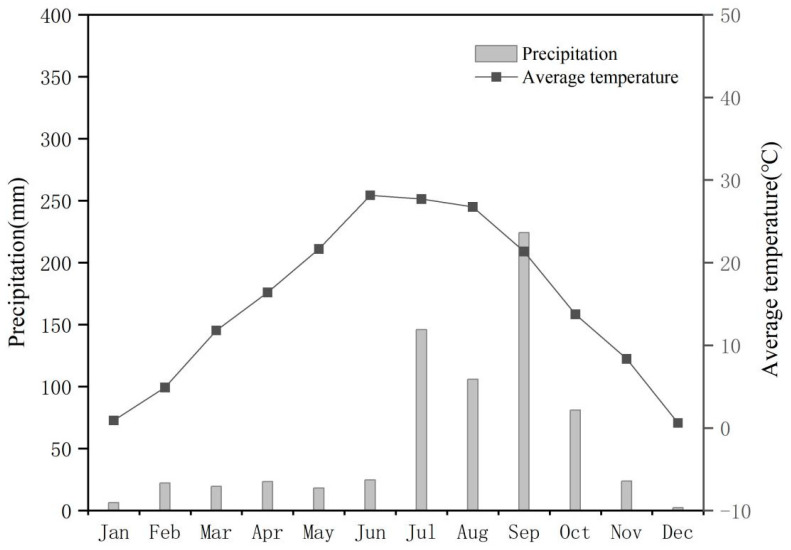
Monthly average meteorological data throughout 2021 and 2022.

**Figure 2 plants-13-00682-f002:**
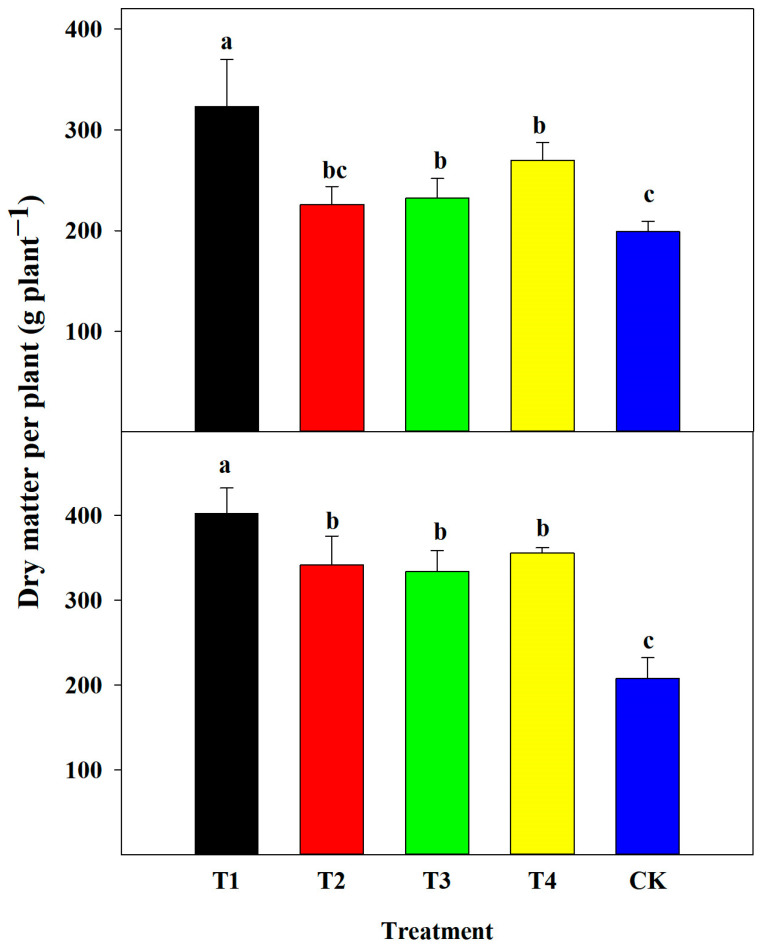
Effect of nutrient deficiency on dry matter per plant at corn harvest stage (R6) in 2021 and 2022. T4, adequate nutrient supply treatment; T3, lack of nitrogen fertilizer treatment; T2, lack of phosphorus fertilizer treatment; T1, lack of potassium fertilizer treatment; CK, no-fertilizer application treatment. Different letters in each column indicate significant differences at *p* < 0.05 (Duncan’s multiple range test).

**Figure 3 plants-13-00682-f003:**
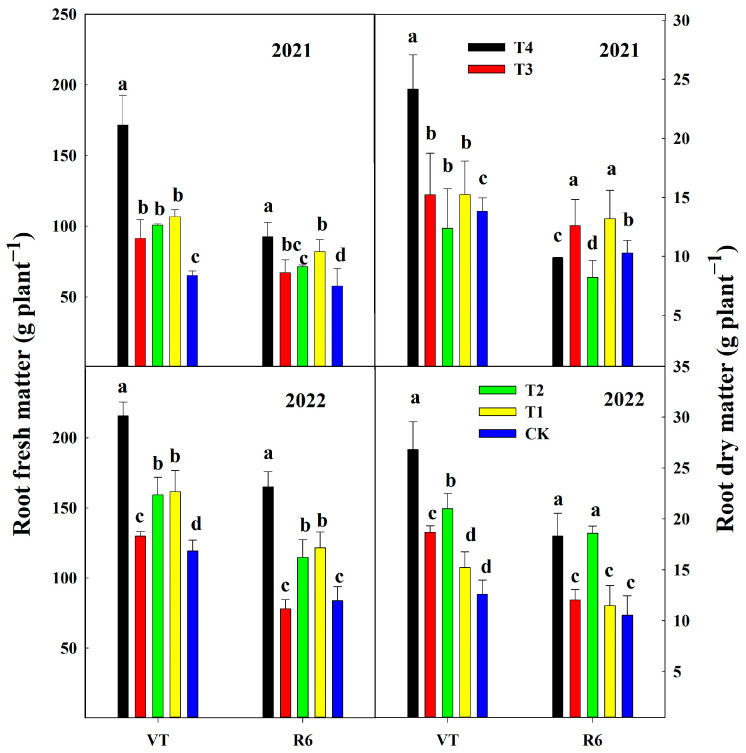
Effect of nutrient deficiency on root fresh matter and root dry matter at corn growth stages (VT and R6) in 2021 and 2022. T4, adequate nutrient supply treatment; T3, lack of nitrogen fertilizer treatment; T2, lack of phosphorus fertilizer treatment; T1, lack of potassium fertilizer treatment; CK, no-fertilizer application treatment. VT, tasseling stage; R6, physiological maturity stage. Different letters in each column indicate significant differences at *p* < 0.05 (Duncan’s multiple range test).

**Figure 4 plants-13-00682-f004:**
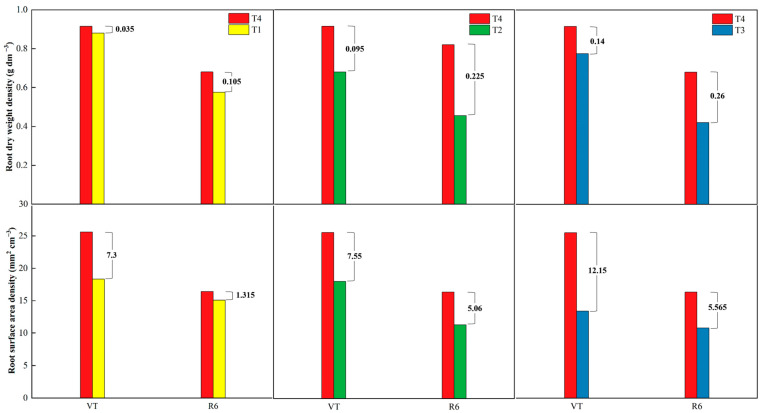
Differences between nutrient-sufficient and nutrient-deficient treatments for root metrics (RDWD and RASD) at corn growth stages. T4, adequate nutrient supply treatment; T3, lack of nitrogen fertilizer treatment; T2, lack of phosphorus fertilizer treatment; T1, lack of potassium fertilizer treatment. VT, tasseling stage; R6, physiological maturity stage.

**Table 1 plants-13-00682-t001:** Properties of the topsoil at the experimental site in 2021 and 2022.

Year	OM (g/kg)	pH	TN (g/kg)	TP (g/kg)	TK (g/kg)	HN (mg/kg)	AP (mg/kg)	AK (mg/kg)
2021	12.94	8.57	0.96	0.86	15.74	54.2	11.3	113
2022	13.06	8.49	0.93	0.75	16.08	63.5	10.9	109

TN: total nitrogen; TP: total phosphorus; TK: total potassium; OM: organic matter; HN: alkaline hydrolysis nitrogen; AP: available phosphorus; AK: rapidly available potassium.

**Table 2 plants-13-00682-t002:** Effect of nutrient deficiency on root length density, root surface area density, and root dry weight density at corn growth stages (VT and R6) in 2021 and 2022.

Year	Soil Layer	Treatment	VT			R6		
			Root Length Density (mm cm^−3^)	Root Surface Area Density (mm^2^ cm^−3^)	Root Dry Weight Density (g dm^−3^)	Root Length Density (mm cm^−3^)	Root Surface Area Density (mm^2^ cm^−3^)	Root dry Weight Density (g dm^−3^)
2021	0–20	T4	9.2 a	16.2 a	0.44 a	2.1 a	6.2 a	0.41 a
		T3	3.1 c	5.9 c	0.27 b	2.2 a	5.2 b	0.38 a
		T2	5.8 b	11.2 b	0.43 a	1.6 b	4.5 bc	0.25 c
		T1	2.9 c	9.8 b	0.15 c	2.3 a	4.9 bc	0.30 b
		CK	3.0 c	5.0 c	0.15 c	1.8 b	4.1 c	0.29 b
	20–40	T4	1.4 a	3.1 a	0.38 bc	0.33 a	0.63 b	0.03 a
		T3	1.5 a	2.2 b	0.47 b	0.32 a	0.77 a	0.04 a
		T2	0.8 b	1.5 b	0.37 bc	0.22 bc	0.53 c	0.02 b
		T1	1.5 a	2.9 a	0.66 a	0.20 c	0.44 c	0.02 b
		CK	0.9 b	2.3 ab	0.34 c	0.27 b	0.53 c	0.03 a
	40–60	T4	2.1 a	4.8 a	0.38 a	0.23 ab	0.48 a	0.02 a
		T3	1.2 b	2.0 b	0.41 a	0.26 a	0.53 a	0.01 a
		T2	1.0 b	1.8 b	00.29 b	0.15 c	0.26 b	0.01 a
		T1	0.9 b	1.8 b	0.42 a	0.19 b	0.35 b	0.02 a
		CK	1.3 b	1.4 b	0.19 b	0.15 c	0.32 b	0.01 a
2022	0–20	T4	5.2 a	16.2 a	0.49 a	8.7 a	21.8 a	0.79 a
		T3	3.3 b	10.4 b	0.3 b	6.0 b	12.1 b	0.34 b
		T2	5.0 a	14.6 a	0.45 a	5.6 b	15.4 b	0.56 a
		T1	4.8 a	15.1 a	0.47 a	8.6 a	21.2 a	0.73 a
		CK	3.3 b	10.5 b	0.28 b	4.0 c	8.0 c	0.31 b
	20–40	T4	2.6 a	5.4 a	0.07 a	1.1 a	2.0 a	0.06 a
		T3	2.0 a	3.8 ab	0.06 a	1.0 a	2.0 a	0.04 a
		T2	2.3 a	4.2 a	0.06 a	0.6 b	1.3 b	0.05 a
		T1	2.0 ab	3.7 bc	0.04 b	0.7 b	2.3 a	0.06 a
		CK	1.6 b	2.7 c	0.06 a	0.5 c	1.1 b	0.03 a
	40–60	T4	2.8 a	5.5 a	0.07 a	0.94 a	1.69 a	0.05 a
		T3	1.3 c	2.6 b	0.04 b	0.52 b	1.07 b	0.03 b
		T2	1.8 b	2.8 b	0.04 b	0.42 c	0.69 c	0.02 b
		T1	2.0 b	3.3 b	0.02 c	0.13 d	0.98 c	0.02 b
		CK	0.5 d	0.7 c	0.05 ab	0.42 c	0.26 d	0.01 c
Year			*	**	**	**	**	*
Soil Layer			**	**	*	**	**	**
Treatment			*	**	NS	NS	NS	*
Year × Soil Layer			NS	*	**	**	**	NS
Year × Treatment			NS	NS	NS	*	NS	*
S L × T			NS	*	NS	*	NS	NS

T4, adequate nutrient supply treatment; T3, lack of nitrogen fertilizer treatment; T2, lack of phosphorus fertilizer treatment; T1, lack of potassium fertilizer treatment; CK, no-fertilizer application treatment. VT, tasseling stage; R6, physiological maturity stage. Different letters in each column indicate significant differences at *p* < 0.05 (Duncan’s multiple range test). NS: not significant; * significant at the 0.05 probability level; ** significant at the 0.01 probability level.

**Table 3 plants-13-00682-t003:** Effect of nutrient deficiency on the root diameter and root length at corn growth stages (VT and R6) in 2021 and 2022.

Year	Soil Layer	Treatment	VT		R6	
			Root Diameter (cm)	Root Length (m)	Root Diameter (cm)	Root Length (m)
2021	0–20	T4	9.2 a	269.4 a	5.4 a	63.1 a
		T3	3.5 b	90.1 c	5.0 a	63.3 a
		T2	7.6 a	171.8 b	4.2 b	46.5 b
		T1	3.4 b	148.6 b	5.1 a	49.7 b
		CK	4.2 b	84.4 c	4.2 b	54.6 b
	20–40	T4	1.8 a	49.1 a	1.1 a	9.3 a
		T3	1.7 ab	43.6 a	0.8 b	9.7 a
		T2	1.4 bc	33.9 b	0.5 c	6.4 b
		T1	1.4 bc	34.5 b	0.4 c	5.9 b
		CK	1.2 c	35.0 b	0.6 bc	6.5 b
	40–60	T4	1.53 a	42.8 a	0.4 a	6.7 ab
		T3	1.53 a	35.6 a	0.4 a	7.5 a
		T2	1.51 a	30.2 b	0.2 a	4.5 c
		T1	1.4 a	38.9 a	0.3 a	5.5 b
		CK	1.5 a	26.4 b	0.4 a	4.5 c
2022	0–20	T4	6.9 a	159.2 a	7.7 a	256.5 a
		T3	4.6 bc	107.9 c	3.9 bc	176.4 c
		T2	6.0 ab	148.2 ab	6.1 a	163.8 c
		T1	6.0 ab	142.2 b	7.6 a	215.8 b
		CK	4.0 c	97.5 c	3.2 c	114.5 d
	20–40	T4	1.7 a	76.1 a	1.3 a	29.3 a
		T3	1.3 ab	58.6 b	1.1 a	25.1 a
		T2	1.3 ab	67.8 a	0.9 b	18.4 b
		T1	0.9 b	48.4 c	0.9 b	18.9 b
		CK	1.3 ab	45.9 c	0.8 b	18.0 b
	40–60	T4	1.6 a	60.3 a	0.7 a	19.2 a
		T3	0.8 b	37.8 c	0.7 a	15.2 b
		T2	0.8 b	52.2 b	0.4 c	12.4 b
		T1	0.2 c	36.8 c	0.1 c	13.3 b
		CK	1.0 b	38.6 c	0.6 b	4.0 c
Year			NS	**	**	**
Soil Layer			**	**	**	**
Treatment			*	**	**	NS
Year × Soil Layer			NS	**	*	**
Year × Treatment			NS	**	**	*
S L × T			*	**	*	**

T4, adequate nutrient supply treatment; T3, lack of nitrogen fertilizer treatment; T2, lack of phosphorus fertilizer treatment; T1, lack of potassium fertilizer treatment; CK, no-fertilizer application treatment. VT, tasseling stage; R6, physiological maturity stage. Different letters in each column indicate significant differences at *p* < 0.05 (Duncan’s multiple range test). NS: not significant; * significant at the 0.05 probability level; ** significant at the 0.01 probability level.

**Table 4 plants-13-00682-t004:** Effect of nutrient deficiencies on annual yield of treatments, number of grains per ear, thousand-kernel weight, and yield of summer corn in 2021 and 2022.

Year	Treatment	Harvest Ear Number (ears·ha^−1^)	Grains per Ear	1000-Grain Weight (g)	Yield (kg·ha^−1^)	Annual Yield (kg ha^−1^)
2021	T1	52,500 a	538 a	295 a	9693 b	
	T2	51,100 a	530 a	299 a	9387 b	
	T3	52,500 a	512 b	283 b	8862 bc	
	T4	55,300 a	565 a	300 a	10,888 a	
	CK	53,000 a	478 b	282 b	8306 c	
2022	T1	48,000 ab	578 c	316 a	8957 b	21,036 b
	T2	47,700 ab	649 a	320 a	9925 ab	23,723 a
	T3	44,700 ab	598 b	267 b	7221 c	20,157 c
	T4	51,100 a	618 ab	323 a	10,202 a	23,531 a
	CK	43,000 b	614 a	287 b	7653 c	19,452 c
Year		**	**	NS	**	
Treatment		*	*	**	**	
Year × Treatment		NS	*	NS	*	

T4, adequate nutrient supply treatment; T3, lack of nitrogen fertilizer treatment; T2, lack of phosphorus fertilizer treatment; T1, lack of potassium fertilizer treatment; CK, no-fertilizer application treatment. Different letters in each column indicate significant differences at *p* < 0.05 (Duncan’s multiple range test). NS: not significant; * significant at the 0.05 probability level; ** significant at the 0.01 probability level.

**Table 5 plants-13-00682-t005:** Correlation of root properties and yield of summer corn.

	Yield	Root Length	Root Diameter	Root Length Density	Root Surface Area Density	Root Dry Weight Density	Root Fresh Matter	Root Dry Matter	Dry Matter per Plant
Yield	1								
Root length	0.919 **	1							
Root diameter	0.781 *	0.658 *	1						
Root length density	0.824 **	0.989 **	0.695 *	1					
Root surface area density	0.779 *	0.973 **	0.792 **	0.983 **	1				
Root dry weight density	0.723 *	0.858 **	0.932 **	0.872 **	0.935 **	1			
Root fresh matter	0.975 **	0.591	0.545	0.544	0.593	0.616	1		
Root dry matter	0.891 **	0.842 **	0.842 **	0.832 **	0.890 **	0.925 **	0.678 *	1	
dry matter per plant	0.865 *	0.814 **	0.77 **	0.808 **	0.841 **	0.811 **	0.563	0.837 **	1

Significance levels: *, significance at *p* < 0.05; **, significance at *p* < 0.01.

## Data Availability

All data generated or analyzed during this study are included in this published article.

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
