# Peer review of "The Effect of Nutrient Deficiencies on the Annual Yield and Root Growth of Summer Corn in a Double-Cropping System"

_plants, 2024, doi:10.3390/plants13050682_

Round 1
Reviewer 1 Report
Comments and Suggestions for Authors
Dear Authors:
- The status of treatments is unclear! A sufficient or insufficient amount of an element has no meaning.
- The amount of yield and components should be mentioned in the article's abstract.
- The conclusion of the article should be corrected in the abstract.
- The method of referencing should be modified, for example: Hazra K K et al. 2018
- Precipitation and temperature of the region for 4 years
- What were the results of the treatments in 4 years? Has the effect of the year been significant? The results of the analysis of variance should be presented.
- In conclusion, which fertilizer treatment should evaluate the performance and constituents of the plant
Comments on the Quality of English Language
It needs minor revision.
Reviewer 2 Report
Comments and Suggestions for Authors
Please check the manuscript to check the comments, suggestions are given using track change mode.

Reviewer 3 Report
Comments and Suggestions for Authors
Materials and methods need to be enhanced as follow:
Add detailed information about agronomic measures applied to all treatments
Add detailed information about the fertilizer units for K, N, and P added to all treatments (T1, T2..)
Add detailed information about seeding rate: seeds for m-2, distance between rows and between plants
Added details about winter wheat: sowing by hand? Weed control manually?
Root diameter: added information about the instrument used to measure the diameter
Soil volume: indicate which volume of soil was used for the calculation
Discussion:
The authors should explain why they present the data as an average of the two years (means of weather conditions during growing seasons… and so on for all data). It would be more useful to analyze the differences for each year and try to identify any interactions with weather.
Round 2
Reviewer 1 Report
Comments and Suggestions for Authors
Dear Authors:
I have reviewed your responses to my comments. but some issues remained:
- I didn't find the ANOVA table.
- There are no descriptions for VT and R6 in tables and graphs.
- There are no descriptions for treatments in figure 2.
- There are no descriptions for TN, TP, and TK in the below of Table 1.
Comments on the Quality of English Language
Dear Editor:
the manuscripts still need some minor corrections.
Thank you
Reviewer 2 Report
Comments and Suggestions for Authors
The present version of the MS could be considered to be published now.
Author Response
Dear Reviewer:
Thank you for your affirmation and wish you a happy life.
Reviewer 3 Report
Comments and Suggestions for Authors
the manuscript has been proofread and is now suitable for publication.
Author Response
Dear reviewer,
Thank you for your affirmation and have a nice life.
Sincerely,
Fei Gao